# Mental and Behavioral Health Disparities Among Pain-Reliever Misusers: A Cross-Sectional Analysis by Race and Ethnicity

**DOI:** 10.3390/healthcare13212674

**Published:** 2025-10-23

**Authors:** James P. D’Etienne, Sam Abduganiev, Ryan Warrior, Hao Wang

**Affiliations:** 1Integrative Emergency Services, Department of Emergency Medicine, JPS Health Network, Fort Worth, TX 76104, USA; jdetienne@ies.healthcare; 2Department of Emergency Medicine, JPS Health Network, Fort Worth, TX 76104, USA; sabdugani@outlook.com (S.A.); rwarrior@jpshealth.org (R.W.)

**Keywords:** pain-reliever misuse, severe psychological distress, suicidal ideation, socio-demographic factors

## Abstract

**Objectives**: The misuse of pain relievers has been linked to mental and behavioral disorders. This study aims to determine the associations between pain-reliever misuse, severe psychological distress (SPD), suicidal ideation, and difficulties in performing daily activities. Additionally, it seeks to identify the socio-demographic factors associated with pain-reliever misuse across different racial and ethnic groups. **Methods**: This cross-sectional study utilizes data from the 2022 United States National Survey on Drug Use and Health (NSDUH). Participants were categorized into four groups: non-Hispanic White (NHW), non-Hispanic Black (NHB), Hispanic/Latino (Hispanic), and Other (American Indian, Alaska Native, Asian, Native Hawaiian or other Pacific Islanders, and two or more races) groups. Comparisons were made between individuals regarding pain-reliever misuse, socio-demographic characteristics, SPD, suicidal thoughts, and World Health Organization Disability Assessment Schedule (WHODAS) scores, using Rao–Scott Chi-square tests. Stepwise multivariable logistic regression analyses were conducted to identify socio-demographic factors associated with pain-reliever misuse. **Results**: The study included 45,451 participants, with 27,551 (62.00 wt%) identified as NHW, 5186 (11.98 wt%) as NHB, 7795 (17.15 wt%) as Hispanic, and 4919 (8.87 wt%) as other racial and ethnic groups. The rate of pain-reliever misuse was 2.90% among NHWs, 3.40% among NHBs, 3.61% among Hispanics, and 2.05% among individuals of other races and ethnicities (*p* = 0.043). Among those who misused pain relievers, a significantly higher proportion experienced SPD (36.00% vs. 14.05%), suicidal thoughts (15.51% vs. 4.91%), and difficulties in performing daily activities (73.77% vs. 52.84%) compared to those who did not misuse pain relievers (*p* < 0.001). Socio-demographic factors associated with a lower risk of misuse included being female (AOR = 0.80, 95% CI 0.67–0.95, *p* = 0.013), being employed (AOR = 0.66, 95% CI 0.48–0.90, *p* = 0.010), and having a college or higher education (AOR = 0.54, 95% CI 0.37–0.79, *p* = 0.002). **Conclusions**: The prevalence of pain-reliever misuse varies across racial and ethnic groups, with Hispanic individuals demonstrating the highest rates of misuse. Pain-reliever misuse is strongly associated with SPD, suicidal thoughts, and impaired daily functioning. Socio-demographic factors are crucial in predicting the likelihood of pain-reliever misuse. These findings highlight the importance of culturally tailored prevention strategies and public health policies aimed at mitigating misuse, especially among vulnerable populations.

## 1. Introduction

In recent decades, a variety of opioid medications, including OxyContin, codeine, fentanyl, and tramadol, have been widely used as pain relievers in the United States (US), leading to a significant increase in opioid overdoses and related fatalities [1]. The mortality rate associated with opioid-related deaths is reported to be even higher among individuals who misuse these pain relievers [2]. The prevalence of opioid misuse and associated mortality rates varies across different racial and ethnic groups [e.g., non-Hispanic White (NHW), non-Hispanic Black (NHB), or Hispanic/Latino] in the US. Historically, NHW populations have exhibited higher rates of opioid misuse likely due to comorbidities and cognitive impairments [3,4]. However, recent data indicates significant increases among NHB and Hispanic populations, with varying motivations for misuse [5,6]. Therefore, further investigation into the patterns of pain-reliever misuse across NHW, NHB, and Hispanic groups is essential.

Concurrently, pain-reliever misuse has been linked to an increased incidence of mental health disorders [7]. Previous research indicates that a significant number of individuals who misuse pain relievers suffer from severe psychogenic disorders (SPDs) with a higher prevalence of suicide [8,9,10]. Such mental health disorders can impede daily functioning, exacerbating their overall health conditions [11,12]. Due to differences in cultural, spiritual, and living conditions, the manifestation of these mental health disorders varies among different NHW, NHB, and Hispanic groups. However, research focusing on the association between pain-reliever misuse and mental health disorders across NHW, NHB, and Hispanic populations remains insufficient.

Informed by Fundamental Cause Theory, which emphasizes how enduring social conditions such as socioeconomic disadvantage, inadequate access to healthcare, and structural racism shape health disparities, this study explores how these underlying factors may contribute to differential patterns of pain-reliever misuse across racial and ethnic groups [13]. Minority Stress Theory also provides a relevant lens to understand how chronic stress from discrimination, marginalization, and social exclusion among racial and ethnic minorities can elevate the risk of psychological distress, suicidal ideation, and maladaptive coping strategies such as substance misuse [14]. By applying these frameworks, this study moves beyond a purely descriptive approach and seeks to identify social and behavioral mechanisms that underlie observed disparities. These theories support our focus on examining both individual-level sociodemographic factors and broader psychosocial stressors in the analysis of pain-reliever misuse and its mental health correlates across diverse populations.

Risks associated with pain-reliever misuse have been documented, with certain demographic factors, such as advanced age and male gender, being associated with higher misuse rates [15,16]. Conversely, factors such as higher education levels and higher socioeconomic status (SES) are negatively associated with pain-reliever misuse [17,18]. Given the increasing trend of pain-reliever misuse among NHB and Hispanic/Latino populations, it is crucial to further examine the factors contributing to misuse among different racial and ethnic groups.

Identifying vulnerable populations at risk for pain-reliever misuse is crucial for developing targeted interventions that can effectively reduce misuse. By mitigating pain-reliever misuse, it may be possible to decrease the prevalence of mental health disorders and enhance their daily functional activities, thereby improving overall healthcare outcomes. To better understand the relationship between pain-reliever misuse and its mental and behavioral health consequences across NHW, NHB, and Hispanic groups, this study aims to: (1) determine the associations between pain-reliever misuse, SPD, and suicidal ideation; (2) examine the relationship between pain-reliever misuse and routine functional behaviors, such as difficulties in performing daily activities; and (3) identify the socio-demographic factors associated with pain-reliever misuse.

## 2. Methods

### 2.1. Study Design and Setting

This study employed a cross-sectional design using data from the 2022 United States National Survey on Drug Use and Health (NSDUH). The NSDUH is a nationally representative survey conducted by the Substance Abuse and Mental Health Services Administration. It collects data through in-person interviews using computer-assisted techniques to ensure respondent privacy and improve accuracy. The survey captures self-reported information on the use of tobacco, alcohol, illicit drugs, and mental health in the civilian, non-institutionalized population in all 50 U.S. states and the District of Columbia. The data is collected continuously throughout the year and released annually, providing timely and comprehensive insights into substance use trends and behavioral health conditions across diverse populations. As NSDUH data are publicly available and do not include any personal health information, this study was classified as non-human subject research and was therefore exempt from regional institutional review board (IRB) review.

### 2.2. Inclusion and Exclusion Criteria

The study population included all U.S. adults aged 18 and older. We excluded individuals who: (1) had missing data on mental and behavioral outcomes (i.e., SPD, suicidal thoughts, and difficulty performing daily activities), and (2) had missing information on marital status or health conditions. A detailed flow diagram of the study population is presented in Figure 1.

### 2.3. Outcome Measurement

The primary outcome of this study was pain-reliever misuse within the past 12 months, as measured by the variable “PNRNMYR: pain reliever misuse in the past year.” In the NSDUH, “pain reliever misuse in the past year” is operationalized based on respondents’ self-reports of using prescription pain relievers in ways not directed by a physician. The misuse definition includes any of the following: using pain relievers without having a prescription, taking them in larger amounts, more frequently, or for longer than prescribed, or using them in any way other than as directed by a doctor [19]. Additionally, three secondary outcomes were assessed: two mental health disorders (SPD and suicidal thoughts) and one functional behavior indicator (difficulty performing daily activities). SPD was measured using the variable “SPDPSTYR: past year severe psychological disorder indicator”. SPD is operationalized using the Kessler Psychological Distress Scale (K6). This scale comprises six questions about the frequency of symptoms (e.g., feeling nervous, hopeless, restless). Respondents rate how often they experienced each symptom (e.g., “none of the time” to “all of the time”), and scores above a validated cutoff (≥13) indicate SPD [19]. Suicidal thoughts were assessed using the variable “SUICTHNK: seriously think about killing self in the past 12 months”. Suicidal thoughts are assessed by asking respondents whether, in the past 12 months, they seriously thought about trying to kill themselves (i.e., had “serious suicidal ideation”). Difficulty in performing daily activities was measured using the “WHODASSCED: WHODAS total score” variable, derived from the World Health Organization Disability Assessment Schedule (WHODAS). A WHODAS score of “0” indicates no difficulty, while a score between “1–24” indicates varying levels of difficulty in performing daily activities. WHODAS includes an abbreviated scale designed to assess impairments in daily functioning. In this study, respondents were considered to have functional impairment if they reported difficulty in any of the following domains: remembering to complete necessary tasks, maintaining concentration when distractions were present, independently leaving the house and navigating their surroundings, interacting with unfamiliar individuals, participating in social activities, managing household responsibilities, fulfilling work or school obligations, and completing daily tasks in a timely manner. These self-reported challenges reflect key dimensions of cognitive, social, and physical functioning, and align with internationally validated WHODAS criteria for identifying functional limitations.

### 2.4. Other Explanatory Variables

This study focused primarily on pain-reliever misuse across different racial and ethnic groups. The key explanatory variable was race and ethnicity, categorized as non-Hispanic White (NHW), non-Hispanic Black (NHB), Hispanic, and Others (including American Indian, Alaska Native, Asian, Native Hawaiian or Other Pacific Islander, and two or more races). Additional socio-demographic variables included age (18–25, 26–34, 35–49, 50–64, and 65+ years), gender (male and female), marital status (married, widowed, divorced/separated, and single), educational attainment (less than high school, high-school graduate, a college or associate degree, and college graduate or higher), household poverty level (poor, near poor, middle/high income), health insurance coverage (insured vs. uninsured), employment status (no employment, employment, and others), and overall health status (excellent, very good, good, fair/poor). Educational attainments are classified as (1) Less than high school: Did not complete high school; (2) High-school graduate: Received a high-school diploma or equivalent; (3) A college or associate degree: Completed a number of college courses or earned an associate degree; and (4) College graduate or higher: Received a bachelor’s, master’s, or doctoral degree [19]. Household poverty level is categorized using the U.S. Census Bureau’s federal poverty thresholds. Respondents are grouped into: (1) Poor: Annual family income below the federal poverty threshold; (2) Near Poor: Income just above the poverty line, often defined as 100–199% of the threshold; and (3) Middle/High Income: Income levels at or above 200% of the poverty threshold [19]. Overall health status in NSDUH is assessed through the following question: “Would you say your health in general is excellent, very good, good, fair, or poor?” This single-item measure has been widely validated and is predictive of morbidity and mortality [20]. The other employment category includes students, persons keeping house or caring for children full-time, retired or disabled persons, or other persons not in the labor force.

### 2.5. Data Analysis

Participants were categorized into four groups based on race and ethnicity (NHW, NHB, Hispanic, and Others), and the prevalence of pain-reliever misuse was compared across these groups. Secondary outcomes (SPD, suicidal thoughts, and difficulty performing daily activities) were also compared among these groups. Weighted percentages (wt%) were calculated for each variable and compared using the Rao–Scott Chi-square test for categorical variables. Stepwise forward multivariable logistic regression analyses were conducted to identify socio-demographic factors associated with pain-reliever misuse. Initially, univariable logistic regression was performed to examine the association between pain-reliever misuse and race and ethnicity. Subsequently, the model was adjusted for SPD, suicidal thoughts, and difficulty performing daily activities to reassess the association with race and ethnicity. Finally, a multivariable logistic regression model including all socio-demographic variables was used to determine the association with pain-reliever misuse. Adjusted odds ratios (AOR) and unadjusted odds ratios (UOR) with corresponding 95% confidence intervals (CI) and *p*-values were reported. The independent variables included in the logistic regression model were selected based on a combination of empirical evidence from prior literature, theoretical relevance to the study objectives, and data availability within the NSDUH [21,22,23,24]. To ensure the robustness of the multivariable logistic regression models, we conducted diagnostic checks to evaluate multicollinearity among the independent variables. Variance inflation factors (VIFs) were calculated for each predictor, and all values were below 10 (ranging from 1.06 to 6.53), indicating no substantial collinearity. Additionally, McFadden’s pseudo R^2^ was estimated from the model as an approximate measure of model fit, and R^2^ < 0.02 indicates poor model fit, 0.02–0.07 indicates weak-to-moderate fit, and >0.07 shows strong and reasonable model fit. This approach, while limited, is common in survey-based studies when likelihood-based statistics are needed but not available in survey-weighted estimators [25,26]. All analyses, including those using weighted replicates, were performed using STATA version 14.2 (College Station, TX, USA).

### 2.6. Reporting Guideline

This study adhered to the Strengthening the Reporting of Observational Studies in Epidemiology (STROBE) guidelines for reporting observational studies [27].

## 3. Results

A total of 59,069 participants were included in the 2022 NSDUH. We excluded 11,969 individuals who were not adults (i.e., aged 12–17), 1620 individuals with missing information on SPD, suicidal thoughts, and difficulty performing daily activities, and 29 individuals with missing data on marital status and health conditions. This resulted in a final sample of 45,451 individuals, representing a weighted population of 248,561,425. A detailed study flow diagram is provided in Figure 1.

Table 1 presents the general characteristics of the study population by race and ethnicity. Among the 45,451 participants, 27,551 (62.00 wt%) were non-Hispanic White (NHW), 5186 (11.98 wt%) were non-Hispanic Black (NHB), 7795 (17.15 wt%) were Hispanic. The pain-reliever misuse rate was 2.90% among NHWs, 3.40% among NHBs, and 3.61% among Hispanics (*p* = 0.043, Table 1). Within the NHW group, the study included a higher proportion of elderly individuals (26.75% aged 65 years or older), fewer individuals in poverty (9.54%), fewer individuals without insurance coverage (6.14%), and fewer unemployed individuals (2.89%) compared to other racial and ethnic groups (*p* < 0.001, Table 1). Among NHBs, a higher proportion of singles (47.78%) and individuals reporting fair/poor health conditions (18.96%) were observed compared to other groups (*p* < 0.001, Table 1). The Hispanic group had a higher proportion of individuals with less than a high-school education (18.64%) and without insurance coverage (19.58%) compared to other groups (*p* < 0.001, Table 1).

Table 2 presents the outcomes across different racial and ethnic groups. NHWs had a higher prevalence of SPD (15.30%, *p* = 0.0364) and difficulty performing daily activities (57.12%, *p* < 0.001) compared to individuals with other races and ethnicities (NHB, Hispanic, and Others). However, there were no statistically significant differences in suicidal thoughts across the four groups (*p* = 0.2186, Table 2). Among those pain-reliever misusers, a significantly higher proportion had SPD (36.00% vs. 14.05%), suicidal thoughts (15.51% vs. 4.91%), and difficulty performing daily activities (73.77% vs. 52.84%) compared to those who did not misuse pain relievers (*p* < 0.001, Table 2).

To further examine the association between race and ethnicity and pain-reliever misuse, stepwise multivariable logistic regressions were conducted. Univariable logistic regression indicated that NHW, NHB, and Hispanic individuals had higher odds of pain-reliever misuse compared to individuals of other races and ethnicities (Table 3). These associations persisted after adjusting for SPD, suicidal thoughts, and difficulty performing daily activities (Table 3). However, after further adjustment for all socio-demographic factors, the adjusted odds ratio (AOR) for NHWs was 1.38 (95% confidence interval [CI] 1.00–1.90, *p* = 0.049), for NHBs was 1.44 (95% CI 0.98–2.14, *p* = 0.065), and for Hispanics was 1.55 (95% CI 1.06–2.29, *p* = 0.026) compared to individuals of other races and ethnicities (Table 3). These findings suggest that both NHWs and Hispanic individuals had significantly higher odds of pain-reliever misuse. Additionally, SPD, suicidal thoughts, and difficulty performing daily activities were consistently associated with pain-reliever misuse (Table 3). Other factors associated with a higher risk of pain-reliever misuse included being aged 35–49 years (AOR = 2.30, 95% CI 1.68–3.14, *p* < 0.001) and having fair/poor health conditions (AOR = 1.95, 95% CI 1.35–2.81, *p* < 0.001). Factors associated with a lower risk of misuse included being female (AOR = 0.80, 95% CI 0.67–0.95, *p* = 0.013), being employed (AOR = 0.66, 95% CI 0.48–0.90, *p* = 0.010), and having a college or higher education (AOR = 0.54, 95% CI 0.37–0.79, *p* = 0.002, Table 3). Additionally, we assessed the model’s goodness-of-fit using McFadden’s pseudo R^2^ (0.10), which showed acceptable explanatory power for a behavioral health model.

## 4. Discussion

In this study, we found that patterns of pain-reliever misuse varied slightly among different racial and ethnic groups, with Hispanics exhibiting the highest rates of misuse. Individuals who engaged in pain-reliever misuse were also more likely to experience SPD, suicidal thoughts, and difficulties in performing daily activities. These mental health disorders and impaired functional behaviors may negatively impact overall healthcare outcomes. Our findings suggest that NHW, Hispanic individuals, and those reporting fair or poor health conditions have higher odds of pain-reliever misuse, whereas females, employed individuals, and those with college or higher education are less likely to misuse these medications. This study provides an updated examination of pain-reliever misuse across various racial and ethnic groups, explores the relationship between misuse, certain mental health disorders, and functional behaviors, and identifies key socio-demographic factors linked to misuse. Our findings contribute additional evidence to the literature on identifying vulnerable populations based on socio-demographic characteristics. Public health researchers and policymakers can use this information to target these at-risk groups with effective interventions, thereby reducing pain-reliever misuse and ultimately improving healthcare outcomes.

The prevalence of pain-reliever misuse in the U.S. varies according to different reports and populations. Previous studies have shown higher rates of misuse among patients with chronic pain, individuals with alcohol abuse, other illicit drug users, and those with poor socioeconomic status [28,29,30,31]. However, our study specifically examines pain-reliever misuse rates across different racial and ethnic groups, an area that is underrepresented in the literature. While individuals from different racial and ethnic backgrounds can also be categorized into vulnerable groups based on their socioeconomic status and chronic disease conditions, understanding the prevalence of pain-reliever misuse across these groups provides a broader perspective on misuse within various communities. Additionally, our study found that individuals who misuse pain relievers are more likely to experience certain mental and behavioral disorders, including SPD, suicidal thoughts, and difficulty performing daily activities. Although these disorders have been associated with other forms of substance abuse, such as alcohol, the prevalence of suicide is notably higher among those misusing pain relievers, suggesting more severe outcomes [32]. The World Health Organization Disability Assessment Schedule (WHODAS) is a widely recognized and externally validated tool used to assess difficulties in performing daily activities. It has been effectively applied to populations with chronic pain, cognitive impairment, alcoholism, and dementia, making it a reliable measure of functional behavior in this study [33,34,35].

In our study, we found that Hispanic individuals tend to have a relatively high prevalence of pain-reliever misuse. In the past, this high prevalence was primarily observed among NHWs. The reasons behind the rapid increase in misuse among Hispanic individuals remain unclear. Some studies suggest it could be related to heightened stress levels within the Hispanic population, while others point to factors such as lower socioeconomic status, language barriers, low-income levels, and limited educational attainment among Hispanic individuals [36,37,38]. Another study suggested that Hispanics might have a relatively low pain threshold, which could contribute to their increased rate of misuse [39].

While our study identified significant differences in pain-reliever misuse across racial and ethnic groups, we observed that NHB individuals did not exhibit statistically significant odds of misuse in the fully adjusted model. Several factors may contribute to this finding. First, although recent reports indicate a rise in opioid-related mortality among NHB populations, underreporting of pain-reliever misuse or differences in healthcare access and utilization patterns may influence how misuse is captured in survey data [40]. Structural inequities, including medical mistrust, lower prescription rates, and disparities in pain treatment, could also contribute to different patterns of opioid access and usage in NHB communities [41]. Moreover, cultural and familial protective factors, such as stronger social networks or spiritual engagement, may buffer against misuse, even in the presence of psychosocial stressors [42]. Lastly, the nonsignificant association may reflect limitations in statistical power for this subgroup or residual confounding not fully addressed by available covariates. These findings highlight the need for more nuanced, culturally informed research to better understand substance use behaviors in NHB populations.

Additionally, we found that middle-aged adults (i.e., those aged 35–49) are more likely to misuse pain relievers, possibly due to the high stress levels typically experienced by this age group [43]. Conversely, being female, employed, or having a college education or higher serves as protective factors against pain-reliever misuse. This could be attributed to greater financial resources linked to better healthcare outcomes among well-educated individuals, and the drug testing requirements often associated with employment [44,45]. Females with co-occurring mental health disorders may seek medical help sooner, potentially reducing their risk of prolonged misuse compared to males, who are less likely to engage in mental health treatment [46].

This study has several strengths. First, it utilizes data from the 2022 National Survey on Drug Use and Health (NSDUH), a nationally representative survey with a significant sample size, making our findings reflective of the national status of pain-reliever misuse. Second, we conducted a comprehensive analysis focused on pain-reliever misuse across four racial and ethnic groups and explored its associations with common mental and behavioral outcomes, which have been linked to the misuse of alcohol, sedatives, and inhalants in previous research. Our findings confirm that misusing pain relievers can increase the occurrence of these disorders. We also identified racial and ethnic differences, with NHWs being the most susceptible to severe psychological distress (SPD) and impaired functional behavior. Third, our focus on socio-demographic characteristics, rather than other potential factors associated with pain-reliever misuse, provides a foundation for developing community-level interventions.

However, our study has limitations. First, while many mental and behavioral outcomes can be associated with pain-reliever misuse, we focused on only three outcomes, SPD, suicidal thoughts, and difficulty performing daily activities, which limits our understanding of the full consequences of misuse. Second, although pain-reliever misuse might be one of the risks for mental and behavioral outcomes, many other factors can influence these outcomes. As a result, we cannot ascertain whether pain-reliever misuse is an independent risk factor or merely a confounding variable when analyzed alongside other factors. In this study, pain-reliever misuse was treated as a binary variable, which may have limited our ability to capture important nuances such as frequency, intensity, or severity of misuse. This simplification could potentially introduce systematic bias and obscure more granular patterns of substance use behavior. Additionally, this study cannot establish a causal relationship between pain-reliever misuse and mental and behavioral outcomes. While these associations are compelling, it is important to emphasize the limitations imposed by the cross-sectional design of this study. Specifically, the directionality of the relationships between pain-reliever misuse and mental health outcomes cannot be determined. It remains unclear whether individuals misuse pain relievers as a form of self-medication in response to psychological distress or whether misuse leads to such mental health conditions. Both pathways are plausible and supported by prior literature [7,47]. Our findings should thus be interpreted as correlational and hypothesis-generating rather than causal. Third, although we emphasize socio-demographic characteristics associated with pain-reliever misuse, other relevant factors, such as individuals’ chronic disease conditions, chronic pain, or other substance use disorders, were not included in our analysis. Chronic pain is a well-documented driver of both legitimate opioid use and pain-reliever misuse, and it often co-occurs with mental health disorders such as depression and anxiety [48]. Similarly, individuals with multiple chronic conditions may be more vulnerable to both substance misuse and psychological distress [49]. Although the 2022 NSDUH dataset includes certain self-reported indicators of overall health status and daily functional impairment (e.g., WHODAS items), it does not provide validated clinical measures of chronic pain intensity, duration, or comorbidity burden. As a result, the associations observed in this study may be confounded by unmeasured health factors. Therefore, future studies focusing on these additional factors at the community level are necessary to further validate our findings.

## 5. Conclusions

The prevalence of pain-reliever misuse varies across different racial and ethnic groups, with Hispanic individuals being the most frequent misusers. Among pain-reliever misusers, non-Hispanic Whites are particularly associated with a higher prevalence of severe psychological distress, suicidal thoughts, and difficulty performing daily activities. Certain socio-demographic factors, such as being non-Hispanic White, Hispanic, middle-aged, and in poor health, are positively associated with pain-reliever misuse. Conversely, being female, employed, or having a college education or higher is negatively associated with the likelihood of misusing pain relievers. These findings highlight the importance of culturally tailored prevention strategies and public health policies aimed at mitigating misuse, especially among vulnerable populations.

## Figures and Tables

**Figure 1 healthcare-13-02674-f001:**
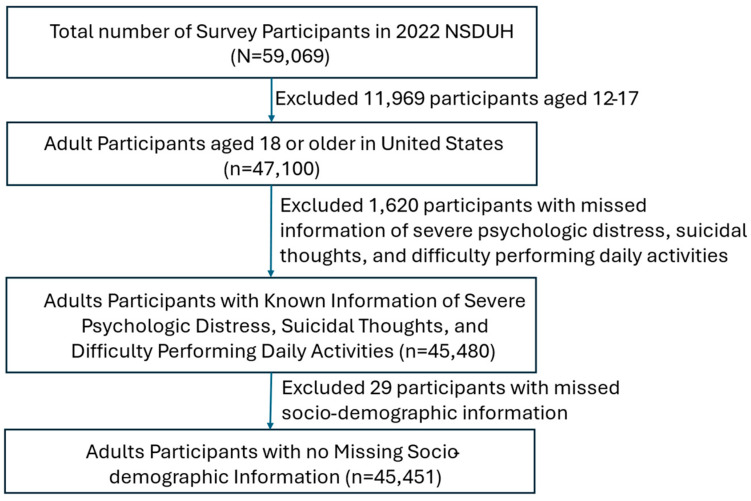
Study Flow Diagram.

**Table 1 healthcare-13-02674-t001:** General Characteristics of the Study Population by Races and Ethnicities.

Variables	NHW 27,551 (62.00)	NHB 5186 (11.98)	Hispanic 7795 (17.15)	Others 4919 (8.87)	*p*
Pain-reliever misuse—n (wt%)					0.043
No	26,715 (97.10)	5000 (96.60)	7522 (96.39)	4781 (97.95)	
Yes	836 (2.90)	186 (3.40)	273 (3.61)	138 (2.05)	
Age—n (wt%)					<0.001
18–25	7203 (11.49)	1722 (15.13)	3117 (18.67)	1729 (16.03)	
26–34	5290 (13.71)	1054 (18.38)	1825 (19.13)	1102 (19.02)	
35–49	7569 (22.26)	1395 (25.57)	1881 (29.30)	1320 (28.43)	
50–64	3539 (25.80)	593 (24.05)	638 (20.70)	433 (20.36)	
65+	3920 (26.75)	422 (16.86)	334 (12.21)	335 (16.16)	
Gender—n (wt%)					0.2555
Male	12,227 (49.12)	2233 (46.32)	3440 (49.62)	2339 (47.73)	
female	15,324 (50.88)	2953 (53.68)	4355 (50.38)	2680 (52.27)	
Marital status—n (wt%)					<0.001
Married	13,166 (53.15)	1179 (28.64)	2685 (44.59)	1901 (55.38)	
Widowed	926 (6.39)	180 (6.95)	112 (3.09)	86 (3.60)	
Divorced/separate	2881 (14.51)	565 (16.64)	660 (12.76)	364 (8.77)	
Single	10,578 (25.95)	3262 (47.78)	4338 (39.56)	2568 (32.26)	
Education—n (wt%)					<0.001
Less than high school	1825 (5.91)	712 (10.97)	1531 (18.64)	475 (11.01)	
High-school graduates	6126 (26.51)	1785 (31.40)	2530 (29.18)	1091 (19.92)	
A college/associate degree	8277 (30.49)	1631 (33.65)	2250 (30.31)	1358 (25.77)	
College or above	11,323 (37.10)	1058 (23.98)	1484 (21.88)	1995 (43.31)	
Poverty level—n (wt%)					<0.001
Poor	3137 (9.54)	1639 (25.11)	2042 (22.17)	922 (14.69)	
Near poor	4585 (16.25)	1325 (25.78)	2219 (26.45)	969 (17.32)	
Middle/high income	19,829 (74.21)	2222 (49.11)	3534 (51.37)	3028 (67.99)	
Insurance—n (wt%)					<0.001
No	1884 (6.14)	672 (10.15)	1615 (19.58)	420 (7.46)	
Yes	25,667 (93.86)	4514 (89.85)	6180 (80.42)	4499 (92.54)	
Employment—n (wt%)					<0.001
No	940 (2.89)	555 (8.22)	583 (6.38)	308 (4.34)	
Yes	18,646 (59.34)	2949 (55.39)	4891 (60.75)	3142 (62.02)	
Others	7965 (37.76)	1682 (36.39)	2321 (32.87)	1469 (33.64)	
Health condition—n (wt%)					<0.001
Excellent	4872 (16.47)	1153 (19.36)	1626 (17.88)	1051 (18.87)	
Very good	10,984 (37.30)	1572 (28.58)	2635 (33.24)	1849 (37.65)	
Good	8577 (31.91)	1701 (33.10)	2485 (33.44)	1397 (30.23)	
Fair/poor	3118 (14.31)	760 (18.96)	1049 (15.44)	622 (13.24)	

Abbreviations: NHW, Non-Hispanic White; NHB, Non-Hispanic Black; wt, weighted; n, number. In Table 1, Chi-square tests were used for categorical variable comparisons.

**Table 2 healthcare-13-02674-t002:** Different Outcome Measurements across Different Racial and Ethnic Groups.

	SPD			Suicidal Thoughts			Difficulty Performing Daily Activities		
Variables—n (wt%)	No	Yes	*p*	No	Yes	*p*	No	Yes	*p*
Race and ethnicity			0.0364			0.2186			<0.001
NHW	21,942 (84.70)	5609 (15.30)		25,518 (94.58)	2033 (5.42)		9783 (42.88)	17,768 (57.12)	
NHB	4369 (87.22)	817 (12.78)		4858 (94.52)	328 (5.48)		2768 (55.09)	2418 (44.91)	
Hispanic	6298 (85.99)	1497 (14.01)		7238 (95.37)	557 (4.63)		3600 (51.82)	4195 (48.18)	
Others	3907 (85.53)	1012 (14.47)		4499 (95.26)	420 (4.74)		1996 (50.25)	2923 (49.75)	
Pain-reliever misuse			<0.001			<0.001			<0.001
No	35,644 (85.95)	8374 (14.05)		40,973 (95.09)	3045 (4.91)		17,806 (47.16)	26,212 (52.84)	
Yes	872 (64.00)	561 (36.00)		1140 (84.49)	293 (15.51)		341 (26.23)	1092 (73.77)	

Abbreviations: SPD, Severe Psychogenic Disorder; NHW, Non-Hispanic White; NHB, Non-Hispanic Black; wt, weighted; n, number. In Table 2, Chi-square tests were used for categorical variable comparisons.

**Table 3 healthcare-13-02674-t003:** Factors Associated with Pain-Reliever Misuse by Stepwise Multivariable Logistic Regressions.

Model-1	Unadjusted Odds Ratios	95% CI	*p*
Race and ethnicity			
Other races and ethnicities (ref)			
NHW	1.42	1.04–1.94	0.028
NHB	1.68	1.15–2.46	0.009
Hispanic	1.78	1.17–2.72	0.008
Model-2	Adjusted Odds Ratios	95% CI	*p*
Race and ethnicity			
Other races and ethnicities (ref)			
NHW	1.36	1.00–1.84	0.049
NHB	1.77	1.22–2.58	0.004
Hispanic	1.83	1.22–2.74	0.004
SPD	2.28	1.84–2.83	<0.001
Suicidal thoughts	1.76	1.31–2.36	<0.001
Difficulty performing daily activities	1.81	1.47–2.23	<0.001
Model-3	Adjusted Odds Ratios	95% CI	*p*
Other races and ethnicities (ref)			
NHW	1.38	1.00–1.90	0.049
NHB	1.44	0.98–2.14	0.065
Hispanic	1.55	1.06–2.29	0.026
SPD	1.96	1.56–2.47	<0.001
Suicidal thoughts	1.68	1.25–2.26	<0.001
Difficulty performing daily activities	1.83	1.49–2.25	<0.001
Age			
18–25 (ref)			
26–34	1.99	1.52–2.60	<0.001
35–49	2.30	1.68–3.14	<0.001
50–64	1.71	1.18–2.47	0.006
65+	1.11	0.73–1.70	0.619
Gender			
Male (ref)			
Female	0.80	0.67–0.95	0.013
Marital status			
Married (ref)			
Widowed	0.66	0.35–0.27	0.211
Divorced/separate	1.26	0.93–1.70	0.136
Single	1.32	1.00–1.75	0.052
Education attainment			
Below high school (ref)			
High-school graduates	0.88	0.67–1.15	0.335
A college/associate degree	0.92	0.70–1.22	0.566
College or above	0.54	0.37–0.79	0.002
Poverty level			
Poor (ref)			
Near poor	0.85	0.65–1.12	0.249
Middle/high income	0.79	0.61–1.02	0.065
Insurance coverage			
No (ref)			
Yes	1.02	0.71–1.46	0.910
Employment			
Employment—No (ref)			
Employment—Yes	0.66	0.48–0.90	0.010
Others	0.68	0.50–0.94	0.021
Health conditions			
Excellent (ref)			
Very good	1.37	0.97–1.95	0.075
Good	1.82	1.22–2.70	0.004
Fair/poor	1.95	1.35–2.81	0.001

## Data Availability

NSDUH data are publicly available at https://www.samhsa.gov/data (accessed on 12 January 2024).

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
