# Peer review of "Mental and Behavioral Health Disparities Among Pain-Reliever Misusers: A Cross-Sectional Analysis by Race and Ethnicity"

_healthcare, 2025, doi:10.3390/healthcare13212674_

Round 1
Reviewer 1 Report
Comments and Suggestions for Authors
The manuscript needs the following clarifications:
- Line 17: “…. 2022 National Survey on Drug Use and Health (NSDUH)” – of which country?
- Line 83 – 84: Please mention the country for a better understanding of the international readers.
- Line 83: “National Survey on Drug Use and Health (NSDUH)” – please mention a little bit about it – for example, frequency of this survey, how data is collected, etc.
- Line 92: “…were aged 12–17 years” – this will not come as exclusion criteria, since it is already mentioned that 18 years or older were included in the study. Exclusion can never be the opposite of the inclusion criteria.
- What was the operational definition of the term “pain reliever misuse in the past year” – please mention with references.
- Line 113 – 118: Better to include the operational definition with references of the following for better clarity for the readers - overall health status (excellent, 117 very good, good, fair/poor), household poverty level (poor, near poor, 115 middle/high income), educational attainment (less than high school, high school graduate, some college or associate degree, and college graduate or higher).
- Line 122: Also mention the operational definition of SPD, suicidal thoughts, and difficulty performing daily activities, for a better understanding of the international readers.
- Table 1: Stub-heading is missing.
- Table 1, 2: Also, better to include the chi-square test statistics along with the p-value (p-value is already written).
- Line 179: Which stepwise model was used and its justification – forward / backward model?
- The output of the logistic regression is missing. Please mention the model fitness, the percentage of variation in the dependent variable that can be explained by the independent variables.
- How were the independent variables chosen in the logistic regression model?
Author Response
Dear Reviewer 1,
1. Line 17: “…. 2022 National Survey on Drug Use and Health (NSDUH)” – of which country?
Response: We revised to “… 2022 United States National Survey on Drug Use and Health (NSDUH).” (Line 17)
2. Line 83 – 84: Please mention the country for a better understanding of the international readers.
Response: We revised to “… 2022 United States National Survey on Drug Use and Health (NSDUH).” (Line 102)
3. Line 83: “National Survey on Drug Use and Health (NSDUH)” – please mention a little bit about it – for example, frequency of this survey, how data is collected, etc.
Response: We appreciate the reviewer’s suggestion and have now included additional information on the National Survey on Drug Use and Health (NSDUH) in the revised manuscript.
“The NSDUH is a nationally representative survey conducted by the Substance Abuse and Mental Health Services Administration. It collects data through in-person interviews using computer-assisted techniques to ensure respondent privacy and improve accuracy. The survey captures self-reported information on the use of tobacco, alcohol, illicit drugs, and mental health in the civilian, non-institutionalized population in all 50 U.S. states and the District of Columbia. The data is collected continuously throughout the year and released annually, providing timely and comprehensive insights into substance use trends and behavioral health conditions across diverse populations.”
We have added this clarification in the Methods section (see Lines 105-111).
4. Line 92: “…were aged 12–17 years” – this will not come as exclusion criteria, since it is already mentioned that 18 years or older were included in the study. Exclusion can never be the opposite of the inclusion criteria.
Response: We thank the reviewer’s valued comments. We have deleted this exclusion criterion (see Lines 117).
5. What was the operational definition of the term “pain reliever misuse in the past year” – please mention with references.
Response: We thank the reviewer’s valued comments. We revised our Methods section with the inclusion of the following:
In the NSDUH, “pain reliever misuse in the past year” is operationalized based on respondents’ self-reports of using prescription pain relievers in ways not directed by a physician. The misuse definition includes any of the following: using pain relievers without having a prescription, taking them in larger amounts, more frequently, or for longer than prescribed, or using them in any way other than as directed by a doctor.
In addition, we added the reference. See Lines 124-128.
6. Line 113 – 118: Better to include the operational definition with references of the following for better clarity for the readers - overall health status (excellent, 117 very good, good, fair/poor), household poverty level (poor, near poor, 115 middle/high income), educational attainment (less than high school, high school graduate, some college or associate degree, and college graduate or higher).
Response: Thank you for your thoughtful suggestion. We have revised the manuscript to include operational definitions of these key sociodemographic measures, as used in the NSDUH, for improved clarity and accessibility for all readers, particularly international audiences.
Educational attainments are classified as 1) Less than high school: Did not complete high school; 2) High school graduate: Received a high school diploma or equivalent; 3) Some college or associate degree: Completed some college courses or earned an associate degree; and 4) College graduate or higher: Received a bachelor’s, master’s, or doctoral degree. [1] Household poverty level is categorized using the U.S. Census Bureau’s federal poverty thresholds. Respondents are grouped into: 1) Poor: Annual family income below the federal poverty threshold; 2) Near Poor: Income just above the poverty line, often defined as 100%–199% of the threshold; and 3) Middle/High Income: Income levels at or above 200% of the poverty threshold. [1] Overall health status in NSDUH is assessed through the following question: “Would you say your health in general is excellent, very good, good, fair, or poor?” This single-item measure has been widely validated and is predictive of morbidity and mortality [2]. We have added these definitions and cited the corresponding references in the revised manuscript (see Lines 165-179).
[1] Substance Abuse and Mental Health Services Administration (SAMHSA). (2022). National Survey on Drug Use and Health: Methodological Resource Book. https://www.samhsa.gov/data/report/nsduh-2022-methodological-resource-book-mrb.
[2] Idler EL and Benyamini Y. Self-rated health and mortality: a review of twenty-seven community studies. J Health Soc Behav, 1997; 38(1):21-37.
7. Line 122: Also mention the operational definition of SPD, suicidal thoughts, and difficulty performing daily activities, for a better understanding of the international readers.
Response: We thank the reviewer’s valued comments. We revised our Methods section with the inclusion of the following:
SPD is operationalized using the Kessler Psychological Distress Scale (K6). This scale comprises six questions about the frequency of symptoms (e.g., feeling nervous, hopeless, restless). Respondents rate how often they experienced each symptom (e.g., “none of the time” to “all of the time”), and scores above a validated cutoff (≥13) indicate SPD. (see Lines 132-136)
Suicidal thoughts are assessed by asking respondents whether, in the past 12 months, they seriously thought about trying to kill themselves (i.e., had “serious suicidal ideation”). (see Lines 137-139)
WHODAS includes an abbreviated scale designed to assess impairments in daily functioning. In this study, respondents were considered to have functional impairment if they reported difficulty in any of the following domains: remembering to complete necessary tasks, maintaining concentration when distractions were present, independently leaving the house and navigating their surroundings, interacting with unfamiliar individuals, participating in social activities, managing household responsibilities, fulfilling work or school obligations, and completing daily tasks in a timely manner. These self-reported challenges reflect key dimensions of cognitive, social, and physical functioning, and align with internationally validated WHODAS criteria for identifying functional limitations. (see Lines 143-152)
We have now added these operational definitions in the revised manuscript so that readers from international or non‑U.S. audiences can better understand these terms in the NSDUH context.
8. Table 1: Stub-heading is missing.
Response: Yes, we added it. (see Table 1)
9. Table 1, 2: Also, better to include the chi-square test statistics along with the p-value (p-value is already written).
Response: We sincerely thank the reviewer for this valuable suggestion. While we agree that including chi-square test statistics alongside the p-values can provide additional context, we have opted not to add a separate column for chi-square statistics in Tables 1 and 2 due to layout and readability considerations. These tables already utilize the full page width, and adding another column may compromise formatting and reduce clarity for key comparative variables. Since the p-values are already presented and serve the primary function of indicating statistical significance, we believe this provides sufficient information for interpretation. Nonetheless, we have included a note in the table footnotes clarifying that chi-square tests were used for categorical variables to enhance transparency. We hope this strikes a reasonable balance between statistical completeness and visual clarity. (see Lines 237-238, and 250-252)
10. Line 179: Which stepwise model was used and its justification – forward / backward model?
Response: We use stepwise forward multivariable logistic regression models. (Line 186)
11. The output of the logistic regression is missing. Please mention the model fitness, the percentage of variation in the dependent variable that can be explained by the independent variables.
Response: We thank the reviewer for this valuable comment. Due to the complex survey design of NSDUH data, including stratification, clustering, and sampling weights, traditional model fit tests such as the Hosmer–Lemeshow test are not appropriate. Instead, McFadden's pseudo R² was estimated from the model as an approximate measure of model fit. This approach, while limited, is common in survey-based studies when likelihood-based statistics are needed but not available in survey-weighted estimators. We revised it in the Methods section of the manuscript with additional references cited. (See Lines 194-204)
12. How were the independent variables chosen in the logistic regression model?
Response: We thank the reviewer for the thoughtful comment. The independent variables included in our multivariable logistic regression model were selected based on prior literature, theoretical relevance, and availability of validated measures within the NSDUH dataset. Specifically, studies have consistently identified socio-demographic factors (e.g., age, gender, race/ethnicity, education, employment), mental health status, and psychosocial functioning as important correlates of pain reliever misuse and other substance use behaviors. To ensure a comprehensive and evidence-based model, we prioritized variables that: 1) Have been previously associated with prescription drug misuse, particularly opioid or pain reliever misuse; 2) Are reliably measured in NSDUH using validated items (e.g., SPD, standardized questions for suicidal thoughts, and WHODAS-based items for functional impairment); and 3) Are relevant for identifying potential disparities across racial and ethnic groups in the context of public health. The stepwise variable selection procedure further refined our model by retaining variables that contributed significantly to the model fit, ensuring parsimony and interpretability while minimizing multicollinearity. We have clarified this rationale in the revised Methods section of the manuscript (see Lines 194-197).
Reviewer 2 Report
Comments and Suggestions for Authors
This manuscript addresses a highly relevant and timely topic at the intersection of the opioid crisis, mental health, and racial/ethnic disparities. The authors have done a great job of structuring the paper logically, utilizing the appropriate statistical methods, and results that are, for the most part, presented in a clear and organized fashion. The study’s focus on identifying vulnerable populations based on socio-demographic characteristics provides actionable information for public health researchers and policymakers.
While there is much to like about this manuscript, there are several concerns that I have:
- The study lacks an explicit theoretical or conceptual framework. The introduction describes trends and associations but does not ground the research in a theory (e.g., minority stress, fundamental cause theory) that could help explain why these disparities in pain reliever misuse might exist across racial and ethnic groups. Without a guiding framework, the analysis remains descriptive rather than explanatory, limiting its contribution to a deeper understanding of the underlying mechanisms driving these health inequities.
- Relatedly, a significant weakness is its reliance on cross-sectional data, which makes it impossible to determine the directionality of the observed relationships. Do mental health challenges like severe psychological distress and suicidal thoughts lead individuals to misuse pain relievers, perhaps as a form of self-medication? Or does the misuse of pain relievers precipitate or exacerbate these mental health conditions? The authors correctly note they cannot establish a causal relationship, but the profound implications of this limitation are undersold. This issue of temporality is a fundamental barrier to interpreting the findings, and it should be discussed more prominently.
- The analysis fails to control for key clinical factors. Most notably, the model does not include measures of chronic pain or chronic disease status, which the authors acknowledge in their limitations. Since chronic pain is a primary driver for both legitimate opioid prescriptions and co-occurring mental health disorders, its absence as a covariate is a critical flaw. The associations reported here could be substantially confounded by the underlying health status of the participants.
- The central term of the study, “pain reliever misuse”, is never explicitly defined. The methods section identifies the variable “PNRNMYR”, but it fails to explain to the reader what behaviors constitute misuse according to the NSDUH (e.g., using without a prescription, using in ways other than prescribed). Providing a clear, operational definition in the introduction or methods is imperative for the reader’s understanding and for the manuscript’s overall clarity. Also, measuring misuse as a simple binary variable (yes/no in the past year) fails to capture the frequency, severity, or context of the misuse, which limits the nuance of the findings.
- There is an inconsistency in the reporting and interpretation of the main findings related to race and ethnicity. In the results section, the authors state that only Hispanic individuals had significantly higher odds of pain reliever misuse, after adjusting for all socio-demographic factors. However, a look at Table 3 shows that both non-Hispanic White individuals and Hispanic individuals had statistically significant higher odds compared to the reference group. The finding for non-Hispanic Black individuals was not significant. This misinterpretation needs to be corrected throughout the manuscript, including the abstract and discussion, to accurately reflect the results of the final adjusted model.
- Several key variables lack sufficient definition or granularity. In Table 1, the “Employment” variable includes a large category labeled “Others”, accounting for over 30% of each racial/ethnic group with no explanation of who this includes (e.g., students, retirees, individuals unable to work). Similarly, the term “non-NHW individuals” is used when describing outcomes, which combines three distinct racial and ethnic groups into one, obscuring potentially important differences between them.
- There is a minor numerical discrepancy between the text and the flow chart regarding the number of participants excluded due to missing socio-demographic information. The text states 27 individuals were excluded for this reason, while Figure 1 indicates 29 were excluded. This should be addressed for consistency.
In sum, this work has clear potential to contribute to the literature. However, the identified weaknesses, ranging from the absence of a theoretical framework and significant methodological limitations (cross-sectional design, risk of omitted variable bias) to a misinterpretation of key findings and a lack of definitional clarity are substantial. The authors need address these concerns before it can be reconsidered for publication.
Author Response
Dear Reviwer 2,
Comment: This manuscript addresses a highly relevant and timely topic at the intersection of the opioid crisis, mental health, and racial/ethnic disparities. The authors have done a great job of structuring the paper logically, utilizing the appropriate statistical methods, and results that are, for the most part, presented in a clear and organized fashion. The study’s focus on identifying vulnerable populations based on socio-demographic characteristics provides actionable information for public health researchers and policymakers.
While there is much to like about this manuscript, there are several concerns that I have:
- The study lacks an explicit theoretical or conceptual framework. The introduction describes trends and associations but does not ground the research in a theory (e.g., minority stress, fundamental cause theory) that could help explain why these disparities in pain reliever misuse might exist across racial and ethnic groups. Without a guiding framework, the analysis remains descriptive rather than explanatory, limiting its contribution to a deeper understanding of the underlying mechanisms driving these health inequities.
Response: We sincerely thank the reviewer for this insightful observation. We agree that integrating a guiding theoretical framework strengthens the explanatory power of our findings. To address this, we have revised the Introduction section to incorporate the Fundamental Cause Theory and the Minority Stress Theory as conceptual lenses to contextualize the observed disparities in pain reliever misuse and its associations with mental health outcomes across racial and ethnic groups. (See Lines 70-82)
- Relatedly, a significant weakness is its reliance on cross-sectional data, which makes it impossible to determine the directionality of the observed relationships. Do mental health challenges like severe psychological distress and suicidal thoughts lead individuals to misuse pain relievers, perhaps as a form of self-medication? Or does the misuse of pain relievers precipitate or exacerbate these mental health conditions? The authors correctly note they cannot establish a causal relationship, but the profound implications of this limitation are undersold. This issue of temporality is a fundamental barrier to interpreting the findings, and it should be discussed more prominently.
Response: We thank the reviewer for this important observation. We fully agree that the use of cross-sectional data in our study presents a fundamental limitation in determining the directionality of the relationships observed between pain reliever misuse, SPD, and suicidal ideation. While our findings highlight strong associations, they cannot establish whether mental health challenges precede pain reliever misuse or whether misuse contributes to the development or worsening of these conditions.
We now acknowledge this limitation more prominently in the Limitation sections of the manuscript. Specifically, we have added language to clarify that the absence of temporal ordering in the cross-sectional design prevents us from inferring causality and that bidirectional or cyclical relationships are plausible. For example, individuals experiencing SPD or suicidal ideation may turn to pain relievers as a coping mechanism, while prolonged misuse of such substances may also exacerbate or even precipitate mental health disorders.
We also emphasize that these findings should be interpreted as hypothesis-generating rather than conclusive. Future studies using longitudinal or prospective cohort designs are urgently needed to disentangle the temporal and potentially causal pathways underlying these associations. Moreover, we agree that the implications of this limitation are far-reaching and have revised our manuscript accordingly to reflect its significance. (See Lines 370-377)
- The analysis fails to control for key clinical factors. Most notably, the model does not include measures of chronic pain or chronic disease status, which the authors acknowledge in their limitations. Since chronic pain is a primary driver for both legitimate opioid prescriptions and co-occurring mental health disorders, its absence as a covariate is a critical flaw. The associations reported here could be substantially confounded by the underlying health status of the participants.
Response: We sincerely thank the reviewer for this thoughtful and constructive comment. We fully agree that the absence of clinical covariates, particularly measures of chronic pain and chronic disease status, is a key limitation of our analysis. Chronic pain is indeed one of the strongest predictors of both legitimate opioid use and misuse, as well as an important factor associated with psychological distress and suicidality. Similarly, chronic disease burden can confound the observed relationships between pain reliever misuse and mental health outcomes, since individuals with multiple chronic conditions are more likely to receive pain medications and experience mental health challenges. Unfortunately, the 2022 NSDUH public-use dataset does not contain detailed, validated measures of chronic pain intensity, duration, or chronic disease diagnoses that would allow for proper adjustment in multivariable models. While certain related indicators (e.g., self-rated health status and functional difficulty from the WHODAS) were included to partially capture health burden, we acknowledge that they cannot fully substitute for clinical pain or disease variables. We have revised the Discussion/Limitations sections to explicitly emphasize that the omission of these variables may lead to residual confounding. (Lines 380-387)
- The central term of the study, “pain reliever misuse”, is never explicitly defined. The methods section identifies the variable “PNRNMYR”, but it fails to explain to the reader what behaviors constitute misuse according to the NSDUH (e.g., using without a prescription, using in ways other than prescribed). Providing a clear, operational definition in the introduction or methods is imperative for the reader’s understanding and for the manuscript’s overall clarity. Also, measuring misuse as a simple binary variable (yes/no in the past year) fails to capture the frequency, severity, or context of the misuse, which limits the nuance of the findings.
Response: We sincerely thank the reviewer for this thoughtful and constructive comment. We revised our Methods section with the inclusion of the following:
In the NSDUH, “pain reliever misuse in the past year” is operationalized based on respondents’ self-reports of using prescription pain relievers in ways not directed by a physician. The misuse definition includes any of the following: using pain relievers without having a prescription, taking them in larger amounts, more frequently, or for longer than prescribed, or using them in any way other than as directed by a doctor. (see Lines 124-128)
In terms of pain reliever misuse, we treated it as a binary variable in our study. Such an approach might minimize essential information, such as the frequency, intensity, or severity of misuse, which could subsequently introduce systematic bias. We realized and emphasized it in the Limitation section (see Lines 365-368).
- There is an inconsistency in the reporting and interpretation of the main findings related to race and ethnicity. In the results section, the authors state that only Hispanic individuals had significantly higher odds of pain reliever misuse, after adjusting for all socio-demographic factors. However, a look at Table 3 shows that both non-Hispanic White individuals and Hispanic individuals had statistically significant higher odds compared to the reference group. The finding for non-Hispanic Black individuals was not significant. This misinterpretation needs to be corrected throughout the manuscript, including the abstract and discussion, to accurately reflect the results of the final adjusted model.
Response: Yes, they are corrected. (see Lines 264, 284)
- Several key variables lack sufficient definition or granularity. In Table 1, the “Employment” variable includes a large category labeled “Others”, accounting for over 30% of each racial/ethnic group with no explanation of who this includes (e.g., students, retirees, individuals unable to work). Similarly, the term “non-NHW individuals” is used when describing outcomes, which combines three distinct racial and ethnic groups into one, obscuring potentially important differences between them.
Response: We revised to include a sufficient definition of “Others” from the employment variable. In addition, we also specified “non-NHW individuals” in the Results section. (See Lines 177-179, 243-244)
- There is a minor numerical discrepancy between the text and the flow chart regarding the number of participants excluded due to missing socio-demographic information. The text states 27 individuals were excluded for this reason, while Figure 1 indicates 29 were excluded. This should be addressed for consistency.
Response: Yes, it is corrected. Thank you. (See Line 213)
In sum, this work has clear potential to contribute to the literature. However, the identified weaknesses, ranging from the absence of a theoretical framework and significant methodological limitations (cross-sectional design, risk of omitted variable bias) to a misinterpretation of key findings and a lack of definitional clarity are substantial. The authors need address these concerns before it can be reconsidered for publication.
Response: Yes, we revised our manuscript accordingly. (see above response to the reviewer’s comments)
Reviewer 3 Report
Comments and Suggestions for Authors
I think the abstract is quite clear in terms of objectives, design and results, and I think it is positive that it presents detailed figures and statistical analyses that allow the relevance of the findings to be appreciated. In my opinion, it is a little long. Furthermore, I think it would be useful to standardise the terminology used to refer to the misuse of analgesics and, in the conclusions, include a sentence about the practical implications of the results, for example, their relevance to prevention or public health policies. Personally, I think that clarifying which groups are referred to by “other racial and ethnic groups” would help to better understand the results.
As for the introduction, it provides a solid context on the use and misuse of opioid analgesics in the United States, as well as their implications for mental health and mortality. In my opinion, clarity could be improved by synthesising some of the information, as I believe that some paragraphs contain too much numerical or historical information. Personally, I think it would be useful to standardise the terminology used to refer to associated mental disorders, as at times ‘SPDs’ are mentioned and at other times more general terms such as depression or anxiety are used, which can cause confusion. I also think it would be helpful to specify more clearly what is meant by ‘diverse racial and ethnic populations’, as this would help to better contextualise the study and its objectives.
I consider that the methods section is very well structured and provides clear information on the study design, inclusion and exclusion criteria, variables, and statistical analyses. However, it would be useful to include a brief justification for the choice of racial and ethnic groups and explain why the socio-demographic variable categories were chosen.
With regard to the results, I believe that this section clearly presents both the descriptive analysis and the statistical associations between the variables, which allows the logic of the study to be followed. The initial characterisation of the population according to race and ethnicity, together with the rates of analgesic misuse and the prevalence of SPD, suicidal thoughts and difficulties in daily activities, is well detailed and facilitates the interpretation of the sample profile. In addition, the stepwise logistic regression analyses allow for the identification of socio-demographic factors associated with analgesic misuse, and the adjusted odds ratios, confidence intervals, and p-values are presented comprehensively, providing transparency on the strength and significance of the observed relationships. In my opinion, the use of models adjusted for SPD, suicidal thoughts, and functional difficulties is relevant and appropriate for understanding the dynamics between the variables.
However, I believe that some aspects could be strengthened. First, although the descriptive analysis is comprehensive, it could be summarised slightly to avoid information overload in tables and text, making it easier to read. In addition, although significant differences between racial and ethnic groups are shown, in my opinion it would be useful to discuss possible causes of these differences and why certain groups, such as NHB, do not reach statistical significance in the final model. I also believe it would be advisable to clarify whether the assumptions of the regression models were evaluated and whether collinearity between variables was explored, especially between socio-demographic and health variables that could influence the results.
As for the conclusions, I believe that this section clearly summarises the main findings of the study on the misuse of analgesics and its associations with sociodemographic and mental health factors. However, I think the conclusions could be strengthened by delving deeper into the practical implications of these findings.
In my opinion, this study provides relevant information on the misuse of analgesics and its relationship with socio-demographic and mental health factors, offering results that may be useful for prevention interventions and public health policies. I consider that the article is well structured and clearly presents the objectives, methods, and results; however, some improvements could be made to strengthen clarity and comprehension. For example, it would be advisable to summarise some of the information in the introduction and results to avoid data overload, standardise the terminology used to refer to analgesic misuse and associated mental disorders, clarify what is meant by “other racial and ethnic populations”, and elaborate on the practical implications of the findings in the conclusions. Incorporating these suggestions would help the article have a greater impact.
Author Response
Dear Reviwer 3,
Comment: I think the abstract is quite clear in terms of objectives, design and results, and I think it is positive that it presents detailed figures and statistical analyses that allow the relevance of the findings to be appreciated. In my opinion, it is a little long. Furthermore, I think it would be useful to standardise the terminology used to refer to the misuse of analgesics and, in the conclusions, include a sentence about the practical implications of the results, for example, their relevance to prevention or public health policies. Personally, I think that clarifying which groups are referred to by “other racial and ethnic groups” would help to better understand the results.
Response: We thank the reviewer’s positive comments. To enhance clarity and consistency, we standardized the terminology throughout the abstract and manuscript to refer to “pain reliever misuse” uniformly. Additionally, we revised the conclusion of the abstract to include a practical implication of the findings, emphasizing their relevance for public health interventions and targeted prevention strategies. Lastly, we acknowledge the need to clarify the term “other racial and ethnic groups” and revise it accordingly. (see Revised abstract)
Comment: As for the introduction, it provides a solid context on the use and misuse of opioid analgesics in the United States, as well as their implications for mental health and mortality. In my opinion, clarity could be improved by synthesising some of the information, as I believe that some paragraphs contain too much numerical or historical information. Personally, I think it would be useful to standardise the terminology used to refer to associated mental disorders, as at times ‘SPDs’ are mentioned and at other times more general terms such as depression or anxiety are used, which can cause confusion. I also think it would be helpful to specify more clearly what is meant by ‘diverse racial and ethnic populations’, as this would help to better contextualise the study and its objectives.
Response: We appreciate the reviewer’s thoughtful and constructive feedback. We agree that improving clarity in the introduction will strengthen the manuscript. We have reviewed the Introduction section and revised several paragraphs to synthesize information more clearly, reducing redundancy and ensuring a more concise presentation of background data and historical context. We recognize that inconsistent terminology may cause confusion. Therefore, we have revised the text to consistently use “severe psychological distress (SPD)” as the primary term when referring to the associated mental health condition measured in this study, with clarification provided at first mention. To improve clarity, we have revised the description of racial and ethnic categories in the introduction to explicitly define “diverse racial and ethnic populations.” These are now consistently referred to as non-Hispanic White (NHW), non-Hispanic Black (NHB), Hispanic/Latino, and other racial/ethnic groups (including Asian, Native American, Pacific Islander, multiracial, and others as defined by NSDUH coding).
These revisions have been incorporated throughout the Introduction section to enhance readability, clarity, and alignment with the study’s objectives. We thank the reviewer for their insightful suggestions, which helped us improve the manuscript. (See revised Introduction section)
Comment: I consider that the methods section is very well structured and provides clear information on the study design, inclusion and exclusion criteria, variables, and statistical analyses. However, it would be useful to include a brief justification for the choice of racial and ethnic groups and explain why the socio-demographic variable categories were chosen.
Response: Yes, we revised the method section with the inclusion of reasons to choose these sociodemographic variables. (See Lines 157-159, 194-197)
Comment: With regard to the results, I believe that this section clearly presents both the descriptive analysis and the statistical associations between the variables, which allows the logic of the study to be followed. The initial characterisation of the population according to race and ethnicity, together with the rates of analgesic misuse and the prevalence of SPD, suicidal thoughts and difficulties in daily activities, is well detailed and facilitates the interpretation of the sample profile. In addition, the stepwise logistic regression analyses allow for the identification of socio-demographic factors associated with analgesic misuse, and the adjusted odds ratios, confidence intervals, and p-values are presented comprehensively, providing transparency on the strength and significance of the observed relationships. In my opinion, the use of models adjusted for SPD, suicidal thoughts, and functional difficulties is relevant and appropriate for understanding the dynamics between the variables.
Response: We appreciate the reviewer’s positive feedback.
Comment: However, I believe that some aspects could be strengthened. First, although the descriptive analysis is comprehensive, it could be summarised slightly to avoid information overload in tables and text, making it easier to read. In addition, although significant differences between racial and ethnic groups are shown, in my opinion it would be useful to discuss possible causes of these differences and why certain groups, such as NHB, do not reach statistical significance in the final model. I also believe it would be advisable to clarify whether the assumptions of the regression models were evaluated and whether collinearity between variables was explored, especially between socio-demographic and health variables that could influence the results.
Response: We thank the reviewer for their thoughtful and constructive feedback. We address each of the points raised below:
We appreciate the suggestion regarding the length and density of the descriptive analysis. To improve readability and avoid information overload, we have carefully reviewed Tables 1 and 2, along with the accompanying text, and have made the following changes: 1) We shortened overly detailed descriptions in the Results section by focusing on the most salient and statistically significant comparisons. 2) Where possible, we grouped related variables or referenced results more generally to reduce redundancy. And 3) We ensured that all table data is directly relevant to the study’s aims and interpretations. We believe that these edits improve the clarity of the presentation while preserving the richness of the descriptive findings. (see Lines 219-235, 241-246)
We agree that discussing the possible causes of racial and ethnic disparities, especially why NHB individuals did not reach statistical significance in the multivariable model is critical to understanding the findings. Accordingly, we have expanded the Discussion section to address: 1) possible structural and social determinants that contribute to differing patterns of pain reliever misuse across racial/ethnic groups; 2) the role of protective cultural factors, underreporting, or differing access to healthcare and prescription medications in NHB populations that may account for their lower adjusted odds; and 3) the impact of statistical power and sample size in subgroup analyses, which may influence whether observed associations reach significance. These additions provide context to interpret both the observed differences and the lack of significance in some groups. (see Lines 322-336)
In terms of model diagnostics and multicollinearity assessment, we thank the reviewer for this important point. We evaluated model diagnostics and multicollinearity among the independent variables. Multicollinearity was assessed using variance inflation factors (VIFs) for all predictors included in the final logistic regression model. All VIF values were below the commonly accepted threshold of 10, indicating no concerning multicollinearity. Although standard model fit indices (e.g., Hosmer-Lemeshow test) are not directly available for survey-weighted logistic regression, we ensured that model convergence and pseudo R² values were acceptable and have included this information in the revised Methods and Results sections. We have added clarifying language in the Methods to indicate that multicollinearity and model assumptions were evaluated during the regression analysis. (see Lines 197-204, 272-274)
Comment: As for the conclusions, I believe that this section clearly summarises the main findings of the study on the misuse of analgesics and its associations with sociodemographic and mental health factors. However, I think the conclusions could be strengthened by delving deeper into the practical implications of these findings.
Response: We appreciate the reviewer’s insightful suggestion. We include practical implications in the conclusion section. (see Lines 398-400)
Comment: In my opinion, this study provides relevant information on the misuse of analgesics and its relationship with socio-demographic and mental health factors, offering results that may be useful for prevention interventions and public health policies. I consider that the article is well structured and clearly presents the objectives, methods, and results; however, some improvements could be made to strengthen clarity and comprehension. For example, it would be advisable to summarise some of the information in the introduction and results to avoid data overload, standardise the terminology used to refer to analgesic misuse and associated mental disorders, clarify what is meant by “other racial and ethnic populations”, and elaborate on the practical implications of the findings in the conclusions. Incorporating these suggestions would help the article have a greater impact.
Response: We thank the reviewer’s valued feedback. As addressed in the previous comments, we made revisions accordingly (see the above responses).
Round 2
Reviewer 2 Report
Comments and Suggestions for Authors
The conclusions section has not been fully corrected. It states: "Certain socio-demographic factors, such as being Hispanic, middle-aged, and in poor health, are positively associated with pain reliever misuse". This conclusion omits the statistically significant finding for NHW individuals. Please revise to include Non-Hispanic White individuals alongside Hispanic individuals as a group with a positive association with pain reliever misuse.
Author Response
Dear Reviewer 2,
Comment: The conclusions section has not been fully corrected. It states: "Certain socio-demographic factors, such as being Hispanic, middle-aged, and in poor health, are positively associated with pain reliever misuse". This conclusion omits the statistically significant finding for NHW individuals. Please revise to include Non-Hispanic White individuals alongside Hispanic individuals as a group with a positive association with pain reliever misuse.
Response: Yes, we revised it to include the NHW in the conclusion. “Certain socio-demographic factors, such as being non-Hispanic White, Hispanic, middle-aged, and in poor health, are positively associated with pain reliever misuse.”